

# A model of multiple tumor marker for lymph node metastasis assessment in colorectal cancer: a retrospective study

Jiangping Fu[1,2,3,*], Mengjie Tu[1,*], Yin Zhang[3], Yan Zhang[4], Jiasi Wang[5], Zhaoping Zeng[1], Jie Li[1] and Fanxin Zeng[1]

[1] Department of Clinical Research Center, Dazhou Central Hospital, Dazhou, Sichuan, China
[2] National Center for International Research of Biological Targeting Diagnosis and Therapy, Guangxi Key Laboratory of Biological Targeting Diagnosis and Therapy Research, Collaborative Innovation Center for Targeting Tumor Diagnosis and Therapy, Guangxi Medical University, Guangxi Zhuang Autonomous Region, Guangxi Zhuang Autonomous Region, China
[3] Department of Oncology, Dazhou Central Hospital, Dazhou, Sichuan, China
[4] Department of Thoracic Oncology, Cancer Center, State Key Laboratory of Biotherapy, West China Hospital, West China Medical School, Sichuan University, Sichuan, China
[5] Department of Clinical Laboratory, Dazhou Central Hospital, Dazhou, Sichuan, China
* These authors contributed equally to this work.

Corresponding authors
Jie Li, 13618272395@163.com
Fanxin Zeng, zengfx@pku.edu.cn

## ABSTRACT

**Background:** Assessment of colorectal cancer (CRC) lymph node metastasis (LNM) is critical to the decision of surgery, prognosis, and therapy strategy. In this study, we aimed to develop and validate a multiple tumor marker nomogram for predicting LNM in CRC patients.

**Methods:** A total of 674 patients who met the inclusion criteria were collected and randomly divided into primary cohort and internal test cohort at a ratio of 7:3. An external test cohort enrolled 178 CRC patients from the West China Hospital. Clinicopathologic variables were obtained from electronic medical records. The least absolute shrinkage and selection operator (LASSO) and interquartile range analysis were carried out for variable dimensionality reduction and feature selection. Multivariate logistic regression analysis was conducted to develop predictive models of LNM. The performance of the established models was evaluated by the receiver operating characteristic (ROC) curve, calibration belt, and clinical usefulness.

**Results:** Based on minimum criteria, 18 potential features were reduced to six predictors by LASSO and interquartile range in the primary cohort. The model demonstrated good discrimination and ROC curve (AUC = 0.721 in the internal test cohort, AUC = 0.758 in the external test cohort) in LNM assessment. Good calibration was shown for the probability of CRC LNM in the internal and external test cohorts. Decision curve analysis illustrated that multi-tumor markers nomogram was clinically useful.

**Conclusions:** The study proposed a reliable nomogram that could be efficiently and conveniently utilized to facilitate the assessment of individually-tailored LNM in patients with CRC, complementing imaging and biopsy tests.

## INTRODUCTION

Colorectal cancer (CRC) is the third most common cause of malignant tumors in the world, with an incidence of 6.1% and a mortality of 9.2% (*Bray et al., 2018*). It is estimated that there will be 3.0 million new cases and 1.5 million CRC related deaths by 2040, which will definitely bring a heavy burden to society (*GLOBOCAN, 2018*). The 5-year survival rate is 90% for localized CRC, 71% for regional disease, but dropping to only 14% when distant metastasis has appeared (*Siegel, Miller & Jemal, 2019*). Lymph node metastasis (LNM), another metastatic mode of CRC, is a main cause of postoperative recurrence and death (*Gunderson et al., 2010*). Therefore, accurate assessment of CRC metastasis, especially LNM, is critical to the decision on surgery, prognosis, and therapy strategy (*Benson et al., 2017*; *Chen & Bilchik, 2006*). Although some histopathological parameters, such as lymphovascular invasion and tumor differentiation, have been reported as the relevant factors of LNM (*Glasgow et al., 2012*), these parameters are not available before surgery. Currently, the LNM of CRC is commonly detected by imaging test, including computed tomography (CT) and magnetic resonance imaging (MRI). However, the imaging modalities have some limitations for assessing LNM in patients with CRC, such as low accuracy (*Brouwer et al., 2018*; *Dighe et al., 2010*). Although previous studies have shown that there are some preoperative prediction nomograms for the LNM (*Huang et al., 2016*; *Qu et al., 2018*), features in these models such as manually extracted features of medical imaging and miRNA expression are unstable and hard-to-get in clinical application. Therefore, developing convenient and accessible noninvasive tumor markers has become an available demand to improve the current methods for assessment of CRC LNM.

Tumor markers, as commonly available biochemical indicators, can be used to evaluate malignant tumor status for assessment of therapeutic effect and prognosis (*Gao et al., 2018*; *Ning et al., 2018*). In recent years, serum tumor markers have played an increasingly important role in the early diagnosis and prognosis of gastrointestinal malignancies (*Duffy et al., 2014*). Previous researches have shown that serum carcinoembryonic antigen (CEA) is a significant indicator of early detection, curative effect, recurrence, and prognostic in patients with CRC (*Song et al., 2012*; *Tarantino et al., 2016*; *Werner et al., 2016*). Furthermore, a study has indicated that the CEA result is comparable to the "gold standard" CT imaging in the evaluation of response for CRC liver metastases to chemotherapy (*de Haas et al., 2010*). Carbohydrate antigen 19-9 (CA19-9) is another commonly used serum tumor marker for CRC in clinic. Several papers have indicated that CA19-9 can be used for postoperative monitoring of CRC (*Chen et al., 2005*; *Kouri, Pyrhonen & Kuusela, 1992*; *Yamashita & Watanabe, 2009*). Some other serum tumor markers, such as alpha-fetoprotein (AFP), carbohydrate antigen 125 (CA125), and carbohydrate antigen 153 (CA153) can also be elevated in CRC (*Dolscheid-Pommerich et al., 2017*; *Huo et al., 2016*;

*Ren et al., 2019*). Although these tumor markers are routine examination items for various tumor patients, the utilization rate of these tumor markers is not high. Multiple tumor markers for assessment of CRC metastasis are still controversial.

In this study, we developed and validated a novel nomogram, combining multiple tumor markers, the ratio of each tumor marker, and clinical risk factors, to predict LNM in patients with CRC. The model more fully utilized the patient's conventional data resources, which made tumor markers' data more valuable. In addition, we evaluated the predictive accuracy and clinical feasibility of the nomogram in a separate internal test and external test cohorts.

## MATERIALS AND METHODS

### Patients

The retrospective analysis was approved by the Medical Ethics Review Board of Dazhou Central Hospital (IRB00000003-19002). And the Medical Ethics Review Board abandoned the need for informed consent from participants in this study. There were 674 CRC patients collected from January 2014 to December 2018, who were histologically diagnosed and have undergone radical surgery with curative intent. These participants were randomly divided into primary cohort and internal test cohort at a ratio of 7:3. An external test cohort enrolled 178 CRC patients from West China Hospital. The inclusion and exclusion criteria for patients were described in the Table S1. The primary cohort contained 471 patients: 290 males and 181 females; mean age, 61.70 ± 11.33 years; range, 20 to 88 years. The internal test cohort incorporated 203 participants: 119 males and 84 females; mean age, 60.27 ± 11.68 years; range, 28 to 89 years. The external test cohort enrolled 178 CRC patients: 119 males and 59 females; mean age, 63.56 ± 13.39 years; range, 27 to 91 years. The clinicopathologic data of baseline, including sex, age, preoperative histologic grade, CEA, AFP, CA125, CA153, and CA19-9, was derived from electronic medical records.

### Index test

The levels of serum CEA, AFP, CA125, CA153, and CA19-9 were measured by the Cobas e601 analyzer when CRC patients were admitted to hospital. All tumor marker assay kits were purchased from Roche (Roche Diagnostics, Switzerland). The specific protocols were in accordance with the kit instructions.

### Definition of groups

LNM group: The diagnosed CRC patients were detected with LNM by imaging application and postoperative histopathologic confirmation within the first period (from admission to surgery).

Non-LNM group: Patients with CRC but disclosed without any LNM and any other metastasis during the first period were included in the non-LNM group.

Metastasis subgroup: The diagnosed CRC patients were detected with significant metastases (LNM or/and distant metastases) by imaging application within the first period.

Non-metastasis subgroup: The CRC patients excluded from metastasis subgroup were included in the non-metastasis subgroup.

## Statistical analysis

Continuous variables were expressed as mean (standard deviation, SD), while categorical variables were shown with count (n) and percentage (%). The Student's $t$ test was applied to compare the age difference between LNM group and non-LNM group. The Pearson Chi-squared test or Fisher's exact test was used to compare sex, tumor stage, and multiple tumor markers level difference between LNM group and non-LNM group. The Kruskal–Wallis test and Chi-squared test were used to compare clinical data among the primary, internal test, and external test cohorts.

The scale function in R software (version R 3.4.2) was used to normalize the data from Dazhou Central Hospital and West China Hospital, respectively. Logistic regression analysis was used to assess the effect of each tumor marker on CRC metastasis in the Dazhou Central Hospital cohort, and the cutoff values of each tumor marker (normalized) were obtained based on Youden index with the "OptimalCutpoints" package in R software. The detailed cutoff value of each tumor marker was shown in Table S2. The AUC, specificity, and sensitivity were estimated on the max Youden index. And the cut off values of each tumor marker was defined as a dichotomic variable in subsequent analysis. When the value of tumor makers below the cut off value, it was defined as "Below$^{\text{cut-off}}$", otherwise it was defined as "Above$^{\text{cut-off}}$". The least absolute shrinkage and selection operator (LASSO) logistic regression method, conducted by 10-fold cross-validation using minimum criteria, was applied to choose the most useful predictive features from different seeds, which were randomly sampled 10 times by stratified sampling in the primary cohort (Fig. S1). The interquartile range (IQR) analysis was used to select predicted indicators from LASSO analyses by different random number seeds, which selected the predictors with cumulative occurrences greater than the median to build a prediction model.

A GiViTI calibration belt was prepared to evaluate the calibration of the multivariable model (Finazzi et al., 2011). The receiver operating characteristic (ROC) curve and area under curve (AUC) were carried out to represent sensitivity and specificity of the model. The optimal critical threshold was determined by the Youden Index. The potential correlation of multivariate with CRC LNM status was first evaluated in the primary cohort and then verified in the internal test and external test cohorts. IBM SPSS statistics 20 and R software (version R 3.4.2) were used for statistical analysis with statistical significance set at $P$-value < 0.05.

## Clinical use

Decision curve analysis was carried out to assess the clinical utility of the multivariable nomogram by quantifying the net benefit at a range of different threshold probabilities in the test data set.

**Table 1 Characteristics of patients in the primary, internal test, and external test cohorts.**

| Characteristic | Primary cohort | | P | Internal test cohort | | P | External test cohort | | P |
|---|---|---|---|---|---|---|---|---|---|
| | LNM (+) | LNM (−) | | LNM (+) | LNM (−) | | LNM (+) | LNM (−) | |
| Age, mean ± SD, years | 60.90 ± 11.32 | 62.28 ± 11.33 | 0.194 | 59.71 ± 12.87 | 60.67 ± 10.77 | 0.563 | 62.03 ± 13.99 | 64.82 ± 12.81 | 0.167 |
| Sex, No (%) | | | 0.480 | | | 0.598 | | | 0.007* |
| Male | 117 (59.69) | 173 (62.91) | | 48 (56.47) | 71 (60.17) | | 45 (56.25) | 74 (75.51) | |
| Female | 79 (40.31) | 102 (37.09) | | 37 (43.53) | 47 (39.83) | | 35 (43.75) | 24 (24.49) | |
| T stage | | | <0.001* | | | 0.001* | | | <0.001* |
| 1 | 24 (12.24) | 112 (40.73) | | 11 (12.94) | 43 (36.44) | | 8 (10.00) | 33 (33.67) | |
| 2 | 131 (66.84) | 125 (45.45) | | 59 (69.41) | 58 (49.15) | | 42 (52.50) | 55 (56.12) | |
| 3 | 41 (20.92) | 38 (13.82) | | 15 (17.65) | 17 (14.41) | | 30 (37.50) | 10 (10.21) | |
| CEA level, No (%) | | | <0.001* | | | 0.010* | | | 0.007* |
| Normal | 94 (47.96) | 182 (66.18) | | 40 (47.06) | 77 (65.25) | | 47 (58.75) | 76 (77.55) | |
| Abnormal | 102 (52.04) | 93 (33.82) | | 45 (52.94) | 41 (34.75) | | 33 (41.25) | 22 (22.45) | |
| AFP level, No (%) | | | 0.201 | | | 0.464 | | | 0.027* |
| Normal | 176 (89.80) | 256 (93.09) | | 76 (89.41) | 109 (92.37) | | 77 (96.25) | 98 (100.00) | |
| Abnormal | 20 (10.20) | 19 (6.91) | | 9 (10.59) | 9 (7.63) | | 3 (3.75) | 0 (0.00) | |
| CA125 level, No (%) | | | 0.142 | | | 0.147 | | | 0.427 |
| Normal | 131 (66.84) | 201 (73.09) | | 57 (67.06) | 90 (76.27) | | 51 (63.75) | 68 (69.39) | |
| Abnormal | 65 (33.16) | 74 (26.91) | | 28 (32.94) | 28 (23.73) | | 29 (36.25) | 30 (30.61) | |
| CA153 level, No (%) | | | 0.081 | | | 0.237 | | | 0.085 |
| Normal | 153 (78.06) | 232 (84.36) | | 70 (82.35) | 89 (75.42) | | 69 (86.25) | 92 (93.88) | |
| Abnormal | 43 (21.94) | 43 (15.64) | | 15 (17.65) | 29 (24.58) | | 11 (13.75) | 6 (6.12) | |
| CA199 level, No (%) | | | <0.001* | | | <0.001* | | | 0.056 |
| Normal | 95 (48.47) | 196 (71.27) | | 39 (45.88) | 89 (75.42) | | 48 (60.00) | 72 (73.47) | |
| Abnormal | 101 (51.53) | 79 (28.73) | | 46 (54.12) | 29 (24.58) | | 32 (40.00) | 26 (26.53) | |

Notes:
* $P$ value < 0.05.
$P$ value is derived from the univariable association analyses between each of the clinicopathologic variables and metastasis status. The tumor stage 1 represents pathological stage T1 and T2, tumor stage 2 represents pathological stage T3, and tumor stage 3 represents pathological stage T4. Abbreviations: CEA, carcinoembryonic antigen; AFP, alpha fetoprotein; CA125, carbohydrate antigen 125; CA153, carbohydrate antigen 153; CA19-9, carbohydrate antigen 19-9; SD, standard deviation.

# RESULTS

## Clinical characteristics

Patient characteristics of the primary, internal test and external test cohorts are shown in Table 1. The positive rates of CRC LNM in the primary, internal test and external test cohorts were 41.61%, 41.87%, and 44.94%, respectively, with no significant differences ($P = 0.736$). There were no statistically significant differences in sex, T stage, CA125, and CA19-9 among the primary cohort, internal test cohort, and external test cohort. The age, CEA, AFP, and CA153 differed significantly among the three cohorts (Table S3). There was also significant difference in age, CEA, AFP, and CA153 among the primary, internal test, and external test cohorts in the subgroup (Table S4).

In the primary cohort, there was a significant difference in both CEA and CA19-9 levels between LNM and non-LNM patients with $P < 0.05$, which was then verified in the internal test cohort or/and the external test cohort (Table 1). As expected, CRC LNM was

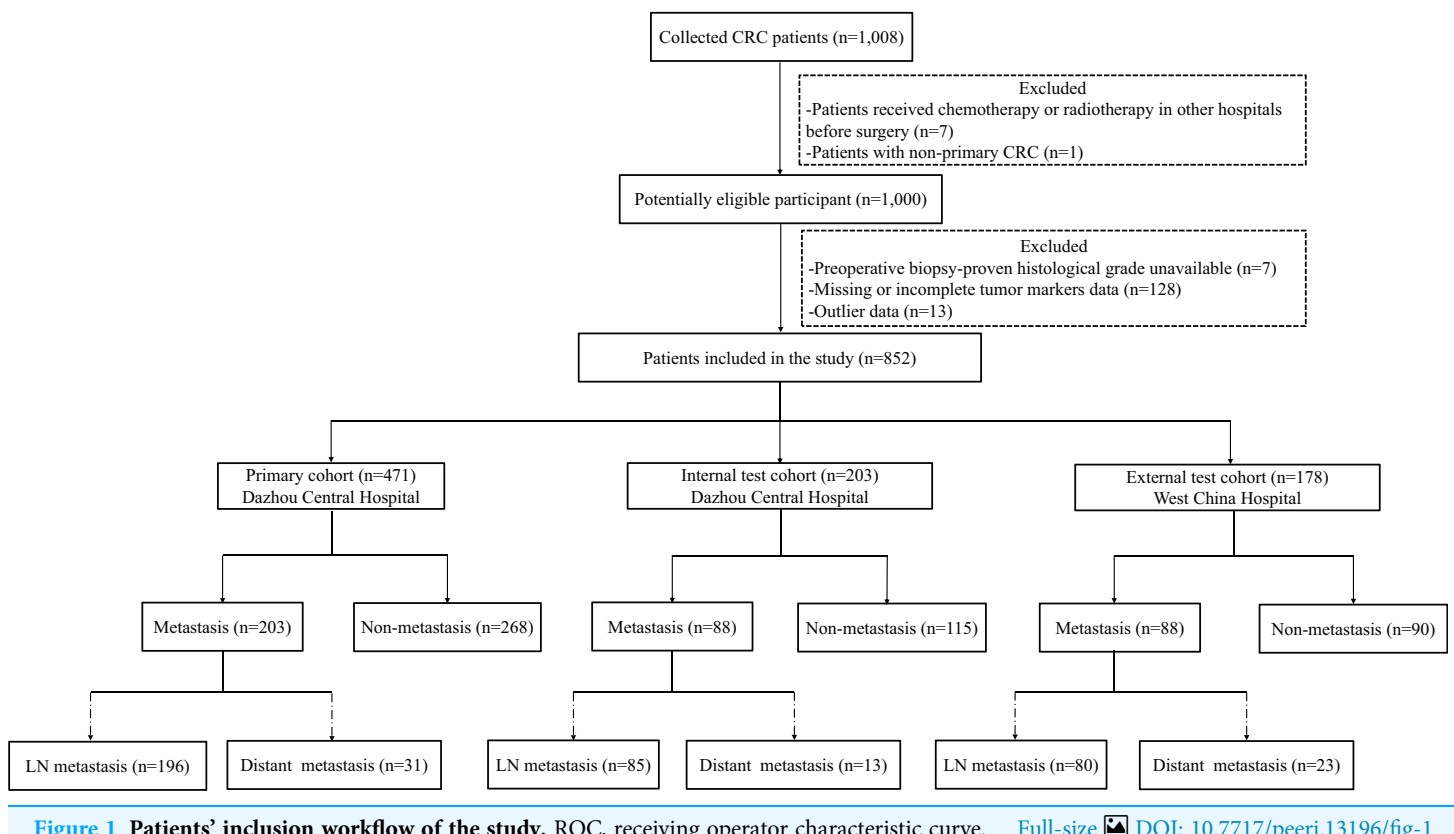

**Figure 1 Patients' inclusion workflow of the study.** ROC, receiving operator characteristic curve.

significantly correlated with the T stage positively in all the primary and internal test and external test cohorts ($P \leq 0.001$) (Table 1). Patients' inclusion workflow of the study is shown in Fig. 1.

## Feature selection and development of an individualized prediction model

In addition to the raw data of tumor markers, the cutoff and ratio of tumor markers (CEA/CA19-9, CEA/CA153, CEA/CA125, CEA/AFP, CA19-9/CA153, CA19-9/CA125, CA19-9/AFP, CA153/CA125, CA153/AFP, CA125/AFP) were included in the predictive analysis. However, the results of univariate analysis showed that the cutoff and ratio of tumor markers had a higher value than the raw data for CRC metastasis (Table S5). Hence, the final 18 features, including the cutoff value, the ratio value, sex, T stage, and patients' age, were used for subsequent analysis. Reduction from 18 features to 2 potential predictors (tumor stage and cutoff_CA19-9) based on 1 standard error of the minimum criteria (1se, right dashed line) and 6 potential predictors (T stage, cutoff_CA19-9, cutoff_CEA, CA19-9/CA125, CEA/CA153, and age) based on minimum criteria (min, left dashed line) by LASSO and IQR analysis in the primary cohort of LNM group (Fig. 2, Fig. S2). Using the same method, 2 (T stage, cutoff_CA19-9) and 5 (T stage, cutoff_CA19-9, CA19-9/CA125, cutoff_CEA, CA19-9/CA153) predictors were found based on 1se and min criteria, respectively, in the metastasis subgroup (Fig. S3). The models for accurately assessing CRC

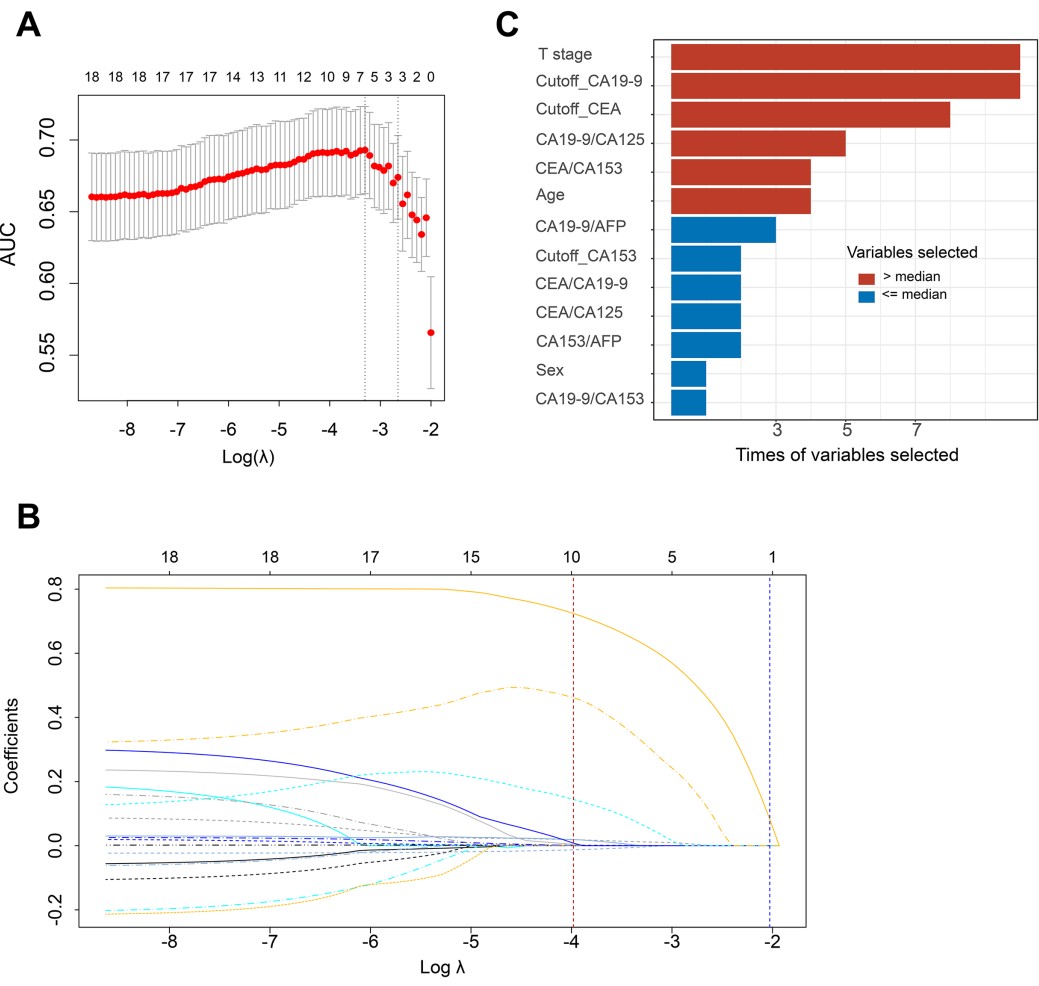

**Figure 2 The texture features of LNM were selected using the LASSO and IQR.** (A) The Tuning parameter (λ) selection in the LASSO model used random stratified sampling 10 times by different seeds (there was only one representative seed, the remaining nine seeds are not shown). The area under the receiver operating characteristic curve was plotted *versus* log (λ). Dotted vertical lines were drawn at the optimal values by using the minimum criteria (min, left dashed line) and the one standard error of the minimum criteria (1se, right dashed line). (B) LASSO coefficient profiles of the 18 texture features. A coefficient profile plot was produced against the log (λ) sequence. Two verticals' lines were drawn at the value selected using min (red line) and 1se (blue line) criteria. (C) According to min results of 10 times LASSO analysis, we chose the variables, of which selected times were greater than the median selected times, to establish the LNM model. LNM, lymph node metastasis; LASSO, least absolute shrinkage and selection operator; IQR, interquartile range.

LNM and metastasis were constructed in the test cohort based on the features selected by 1se and min, respectively.

## Assessment of the nomogram performance

ROC analysis illustrated that the 1se and the min models could reliably differentiate CRC patients with LNM or metastasis from those without metastasis (Fig. 3, Fig. S4). As Fig. 3A shown, there was no significant differences between the 1se model and the min model for predicting LNM ($P = 0.342$) in the internal test cohort. The AUC, specificity, and

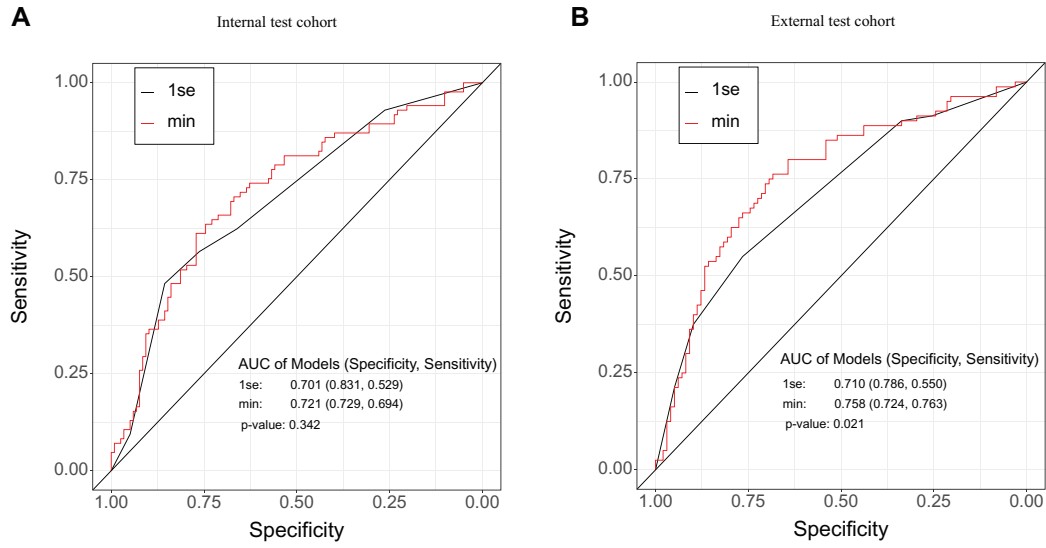

**Figure 3 ROC analysis of LNM assessment model in the internal test and external test cohorts.**
(A) ROC analysis based on the sensitivity and specificity of multivariate, selected by minimum criteria (min) and one standard error of the minimum criteria (1se), for assessing LNM in the internal test cohort. (B) ROC analysis based on the sensitivity and specificity of multivariate, chosen by min and 1se, for assessing the LNM in the external test cohort.

**Table 2 Performance of the LNM prediction models.**

| | Internal test cohort | | External test cohort | |
| --- | --- | --- | --- | --- |
| | 1se model | Min model | 1se model | Min model |
| Specificity (95% CI) | 0.831 [0.669–0.907] | 0.729 [0.534–0.856] | 0.786 [0.337–0.939] | 0.724 [0.571–0.888] |
| Sensitivity (95% CI) | 0.529 [0.400–0.682] | 0.694 [0.518–0.859] | 0.550 [0.350–0.938] | 0.763 [0.550–0.888] |
| Accuracy (95% CI) | 0.700 [0.635–0.759] | 0.709 [0.640–0.768] | 0.674 [0.596–0.736] | 0.736 [0.674–0.792] |
| PPV (95% CI) | 0.685 [0.571–0.804] | 0.644 [0.553–0.754] | 0.672 [0.529–0.839] | 0.693 [0.611–0.818] |
| NPV (95% CI) | 0.707 [0.660–0.769] | 0.765 [0.699–0.853] | 0.679 [0.621–0.878] | 0.784 [0.699–0.871] |
| AUC (95% CI) | 0.701 [0.631–0.772] | 0.721 [0.649–0.793] | 0.710 [0.637–0.783] | 0.758 [0.686–0.831] |

**Note:**
LNM, lymph node metastasis; Min, minimum criteria; 1se, 1 standard error of the minimum criteria.

sensitivity of the 1se model were 0.701, 0.831, and 0.529, respectively; while the AUC, specificity, and sensitivity of the min model were 0.721, 0.729, and 0.694, respectively (Fig. 3A, Table 2). However, in the external test cohort, there was a significant difference between the 1se model and the min model to assess LNM ($P = 0.021$), which AUC, specificity, and sensitivity of the 1se model reached 0.710, 0.786, and 0.550, respectively; and the AUC, specificity, and sensitivity of the min model reached 0.758, 0.724, and 0.763, respectively (Fig. 3B, Table 2). In the metastasis subgroup, the 1se model and the min model had significant difference in the internal test cohort ($P = 0.006$), but there was no significant difference in the external test cohort ($P = 0.07$) (Figs. S4A, S4B). In the internal test cohort of metastasis subgroup, the AUC, specificity, and sensitivity reached 0.724, 0.800, and 0.568, respectively, based on 1se criteria; while the AUC, specificity, and

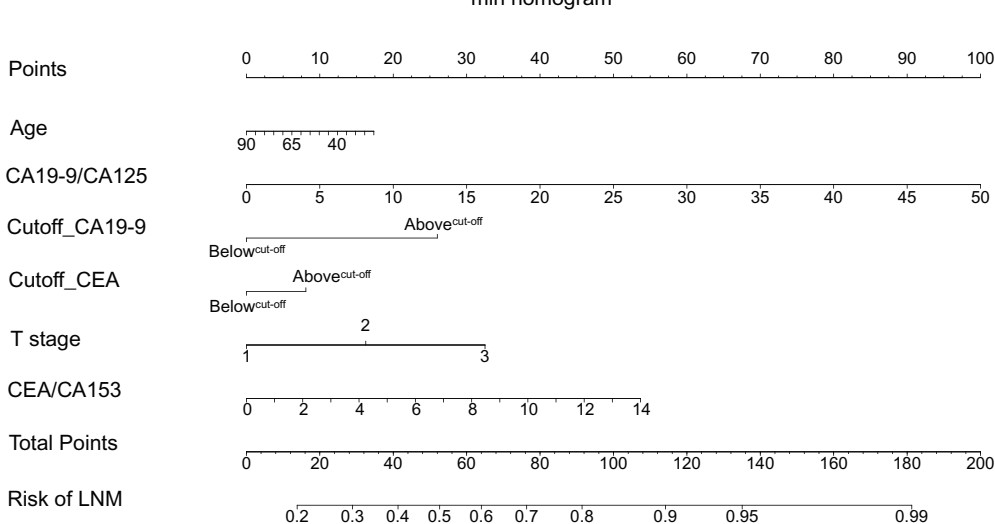

**Figure 4  The nomogram for LNM.** The nomogram was developed in the primary cohort, based on min criteria, with age, CA19-9/CA125, cutoff_CA19-9, cutoff_CEA, tumor stage, and CEA/CA153. CA, carbohydrate antigen; CEA, carcinoembryonic antigen. Nomogram read guidance: Score each variable according to its value level (represented by Points line on the nomogram), and then add the scores of each variable to get the total score (represented by Total points line on the nomogram). The probability corresponding to the vertical line of the total score is the probability of LNM in CRC patients, and the higher the score, the higher the probability of LNM.

sensitivity reached 0.760, 0.704, and 0.727, respectively, based on min criteria (Fig. S4A, Table S6). In the external test cohort of metastasis subgroup, the AUC, specificity, and sensitivity of the min model were 0.740, 0.800, and 0.648, respectively; while the AUC, specificity, and sensitivity of the 1se model were 0.704, 0.789, and 0.545, respectively (Fig. S4B, Table S6). As a result, the models were presented as the nomograms (Fig. 4, Figs. S4C, S4D).

The 80% and 95% confidence intervals of the GiViTI calibration belt of the the min multivariable nomograms for the probability of CRC LNM and metastasis in the internal test cohort did not cross the 45°diagonal bisector, and corresponding $P = 0.562$ and $P = 0.108$ (Fig. 5A, Fig. S5A), which suggested good consistency between prediction and observation of the models in the internal test cohort. Good calibration was also shown for the probability of CRC LNM and metastasis in the respective external test cohorts corresponding with $P = 0.489$ and $P = 0.875$ (Fig. 5B, Fig. S5B).

## Clinical application

The decision curve analysis of the multivariate nomograms based on 1se criteria and min criteria is presented in Fig. 5C and Fig. S5C. The decision curve demonstrated that if the threshold probability of either the doctor or the patient was >20%, using the multivariate nomograms to predict CRC LNM and metastasis could increase more net benefit than either the treat-all-patients plan or the treat-none plan (Fig. 5C, Fig. S5C). The cut-off point of the min model for CRC LNM assessment that we calculated in primary

Peer

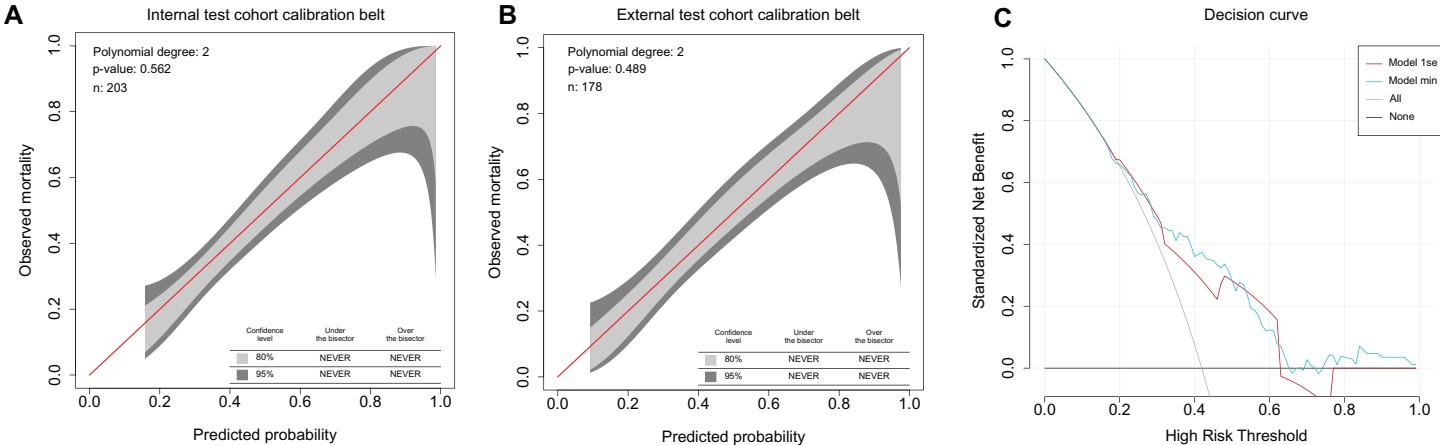

**Figure 5 Calibration belt and decision curve analysis of the multivariate nomogram for LNM.** (A) Calibration belt of the min multivariate nomogram in the internal test cohort. (B) Calibration belt of the min multivariate nomogram in the external test cohort. (C) The vertical-axis measures the net benefit. The red line represents the model based on 1se model. The blue line represents the model based on min model. The gray line represents the assumption that all patients have LNM. Thin black line represents the assumption that no patients have LNM.

cohort was 0.360. Within these ranges, the net benefits were comparable on the basis of the 1se nomogram and the min nomogram with several overlaps.

## DISCUSSION

Tumor markers have a long history and are commonly used to monitor the progression of cancer after curative treatment. CA19-9 and CA724 have been reported vital indicators of disease recurrence and overall survival in CRC (*Zheng et al., 2001*). Besides, study has revealed that preoperative serum CEA is positively correlated to lymph node metastasis and pTNM staging, with positive rates of CEA 24%, 44%, 56% and 87% from stage I to stage IV, respectively (*Gao et al., 2018*). The value of tumor markers for predicting preoperative tumor metastasis was ambiguous. However, most previous studies mainly focused on reference values of tumor markers from test kit and/or the single biomarker, the use of these serum biomarkers for LNM and metastasis assessment in CRC remains to explore. Therefore, in the present study, we developed and verified a diagnostic nomogram based on multiple tumor markers for auxiliary prediction of LNM in CRC patients. The easy-to-use nomograms, including multiple tumor markers status and clinical risk factors, were conducive to the assessment of CRC LNM. Our results showed that the min model incorporated with age, T stage, CA19-9/CA125 value, CEA/CA153 value, cutoff_CA19-9 and cutoff_CEA was reliable in assessing CRC patients with or without LNM.

Although model with more features may reduce biases, it can possibly result in less accurate predictions and affect the efficiency of the estimation procedure. So it is desirable to select the most important variables. There are many conventional methods for feature selection, such as LASSO, ridge regression and ordinary least squares. But previous experimental results showed that the LASSO works better than the other methods by shrinking the coefficients exactly to zero (*Muthukrishnan, 2016*). Besides, LASSO has been

successfully applied to feature selection and model establishment in many existing reports. Hence, we choose the LASSO analysis as the method to the feature selection. In this study, potential predictors were screened from 18 candidate features using the LASSO method. To reduce the sampling error, we randomly stratified sampling 10 times by different seeds. Each sampling applied 10 folds LASSO regression. Finally, we selected 2 and 6 predictors to develop 1se and min models for LNM prediction, respectively. In terms of these tumor markers, CEA and CA19-9 were found more associated with LNM, which in line with currently published studies (*Huang et al., 2016*; *Li et al., 2020*). Interestingly, CA19-9-related variables were more likely to be screened in different seeds than CEA-related variables in our study. In the clinical-radiomics nomogram model for LNM prediction in CRC developed by *Li et al. (2020)*, CA19-9 has greater weight of feature coefficients than CEA. *Huang et al. (2020)* have also revealed that CA19-9 has higher AUC than CEA for LNM assessment in gastric cancer. These findings parallel the results displayed in Tables S4 of our study. In addition, the ratios and cut-off value of tumor markers were screened to develop predictive models, which could achieve AUC 0.721 and 0.758 in internal test cohort and external test cohort, respectively. Our model showed better performance than the clinical features model in previous study (Training cohort of previous study: AUC = 0.7127, Validation cohort of previous study: AUC = 0.7075) (*Li et al., 2020*). These results suggested that the ratios and cut-off value of tumor markers might have more potential than tumor marker value to evaluate LNM and metastasis in CRC.

The nomogram as a statistical model for optimizing the accuracy of individual prediction has an advantage of visualization. Nomogram to evaluate tumor metastasis can assist clinicians in determining the optimal individual treatment options for patients to achieve greater clinical benefit (*Balachandran et al., 2015*; *Kim et al., 2014*; *Thompson et al., 2014*). The traditional detection methods (CT, PET, MRI) of preoperative metastasis are often limited by financial burdens, radiation and low sensitivity. In this study, we developed the models containing multiple tumor marker features, T stage, and age, which were readily available. Furtherly, the regression nomogram which visualized from our model is convenient and accessible in clinical application. Therefore, these nomograms are expected to be a new auxiliary method to guide the treatment of CRC patients, complementing imaging and biopsy tests.

Based on the ROC results, the 1se and min model could successfully distinguish the CRC LNM. There was a significant difference between the ROC of the 1se model and that of the min model in the external test cohort (1se model: AUC = 0.710, min model: AUC = 0.758, $P = 0.021$), which indicated that the min model was more accurate than 1se in assessing LNM. Calibration belt analysis has demonstrated that the min models was available. In addition, even in the 1se model and min model for LNM or metastasis assessment, T stage were the preserved risk factors. T stage is considered as a credible category of the tumor size and depth of tumor invasion. Studies have reported that patients with advanced T stage (T3/T4) had poor prognosis (*Engstrand et al., 2018*; *Sasaki et al., 2016*). And *Wu et al. (2020)* demonstrated that patients with early T stage (T1/T2) had less

LNM than patients in advanced T stage. Our results are consistent with existed findings that CRC patients with advanced T stage are more likely to develop LNM.

The most indispensable argument for using nomograms in the clinic is focused on whether nomogram-assisted decision in surveillance could improve patient treatment and care. However, current methods of assessing the performance of predictive nomograms, such as the calibration belt and AUC, could not acquire the clinical consequences of a specific level of distinction or degree of miscalibration (*Collins et al., 2015*; *Localio & Goodman, 2012*; *Van Calster & Vickers, 2015*). Therefore, decision analysis curves were used to estimate the clinical usefulness of the assessment nomograms on the basis of threshold probability (*Balachandran et al., 2015*; *Vickers et al., 2008*; *Vickers & Elkin, 2006*). It is suggested by the decision curves that if the threshold probability of a doctor or patient is >20%, using the multivariate nomograms to predict CRC LNM will add more net benefit than either the treat-all-patients plan or the treat-none plan.

This study had several limitations. First, since the participants came from the Sichuan province and belonged to the same race as a single-center retrospective study, the lack of generalizability was the main limitation of this study. Second, the tumor markers related to tumor progression, such as CA72-4, was not contained in the study. Therefore, it is necessary to further validate our results through prospective studies of different ethnic groups.

## CONCLUSIONS

Our study proposed a reliable prediction nomogram, which could be efficient and convenient for facilitating the preoperative individually-tailored metastasis prediction in patients with CRC, complementing imaging and biopsy tests.

### Funding

This study was supported by the National Natural Science Foundation of China (81902861), the Innovative Scientific Research Project of Medical Youth in Sichuan Province (Q20073), the Scientific Research Fund of Technology Bureau in Dazhou (19YYJC0010) and the Scientific Fund of Health Commission of Sichuan Province (18PJ040). The funders had no role in study design, data collection and analysis, decision to publish, or preparation of the manuscript.

### Grant Disclosures

The following grant information was disclosed by the authors:
National Natural Science Foundation of China: 81902861.
Innovative Scientific Research Project of Medical Youth in Sichuan Province: Q20073.
Scientific Research Fund of Technology Bureau in Dazhou: 19YYJC0010.
Health Commission of Sichuan Province: 18PJ040.

## Competing Interests

The authors declare that they have no competing interests.

## Author Contributions

- Jiangping Fu performed the experiments, authored or reviewed drafts of the paper, and approved the final draft.
- Mengjie Tu performed the experiments, authored or reviewed drafts of the paper, and approved the final draft.
- Yin Zhang performed the experiments, authored or reviewed drafts of the paper, and approved the final draft.
- Yan Zhang analyzed the data, authored or reviewed drafts of the paper, and approved the final draft.
- Jiasi Wang performed the experiments, analyzed the data, prepared figures and/or tables, authored or reviewed drafts of the paper, and approved the final draft.
- Zhaoping Zeng analyzed the data, prepared figures and/or tables, authored or reviewed drafts of the paper, and approved the final draft.
- Jie Li conceived and designed the experiments, performed the experiments, prepared figures and/or tables, authored or reviewed drafts of the paper, and approved the final draft.
- Fanxin Zeng conceived and designed the experiments, authored or reviewed drafts of the paper, and approved the final draft.

## Human Ethics

The following information was supplied relating to ethical approvals (*i.e.*, approving body and any reference numbers):

Medical Ethics Review Board of Dazhou Central Hospital.

## Data Availability

The raw measurements are available in the Supplemental Files.

## Supplemental Information

Supplemental information for this article can be found online at http://dx.doi.org/10.7717/peerj.13196#supplemental-information.

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
