# Peer review of "A model of multiple tumor marker for lymph node metastasis assessment in colorectal cancer: a retrospective study"

_PeerJ, doi:10.7717/peerj.13196_

## Round 0.1 · original submission · Major Revisions

Please follow the Reviewers' suggestions.

·

Basic reporting

Line 68 - 69: a very brief description of how those previously developed prediction nomograms perform, what they lack and how the authors propose to fill the gap would be helpful here.

Line197 -199: It would help readers if all the ROC plots could be in the same place. For example, moving Sig S4 plots to combine with Fig3.

Line 206: for the nomograms, and all the results in general, why are some figures chosen to be put in the main figure whereas others were in the supplementary? It would be much helpful if the authors could explain the logic and rationale and comparisons between models.

Experimental design

Line 206: despite nomogram being commonly used to visualize models, I recommend authors give basic guidance on how to read a nomogram. What does the length of each variable indicate, how to calculate total points. These could be explained in the legends, methods, or result section.

Line 207: again how do the authors interpret results from the calibration belt? It would be very helpful for authors to very briefly guide readers, who may not be experts in the immediate field to interpret the plots. It would be good to not assume readers understand all the statistical terminology and plots.

Line 235: for discussion, can the authors explain why LASSO regression was chosen for feature selection and model building in the first place? Why is Lasso the most suitable tool for identifying multiple markers?

Validity of the findings

Line 228, I would like to see a brief comparison of this new model the author developed with previous models referenced in the introduction.

Line248: When the authors made the claim that their model showed better performance than the clinical features model in the previous study, they did not show results from the previous study. Thus this conclusion is unjustified as-is.

Line 257-258: Please explain why the regression nomogram is more convenient and accessible by comparison.

Reviewer 2 ·

Basic reporting

Authors proposed a new nomogram for lymph node metastasis assessment in colorectal cancer. They used LASSO algorithm to select patterns, and then combining them in a logistic regression model. The methodology seems to be adequate but, to further process the manuscript, there are issues in material and methods that must be clarify.

Experimental design

I have some concerns:
- Authors used an external validation cohort to asses the predictive abliity of their model, this is the most appropriate process to validate a model, but then it is not understood why they have divided the original database in percentage 7:3, a 10-fold cross-validation should be the best way to internally validated their prediction model. Choosing cutoff points using the 70% of database, and training the multivariate model with the rest 30% would have been understandable but it is not the procedure that the authors have followed
- Authors declare that "Logistic regression analysis was used to assess the effect of each tumor marker on CRC metastasis in the primary cohort, and the cutoff values of each tumor marker were obtained." This point must be clarify, it is not clear if they used the youden index to define a dichotomic variable for each marker, and this categorical variables (as appears in Table 1) are used in the multivariate model, or you are using the cutoff as a continuous variable in the prediction. In a case you used a dichotomic variable, as it seems in the nomogram, you must define these new variables as categorical CAmarker or similar, to an easy interpretation of the results. In any case, to dichotimize markers provides you a more adjusted predictions to the development cohort, but rest a general applicability of the model in different populations.
- In the results authors provide information about AUC, sensitivity and specificity. I suppose that sens and spec are estimated on the max Youden index. It is important to specify this in material and methods, and also to provide the cutoff point. Authors must verified that the Youden cutoff point of the models are in the range of values where the net benefit is superior in the decision curve.

Validity of the findings

The conclusions are derived from a complete validation process based on discrimination, calibration and clinical utlity, thus, avoiding the issues that authors must clarify about the experimental design, their conclusions are valid.

---

## Round 0.2 · accepted · Accept

The authors properly performed the revisions.

Reviewer 2 ·

Basic reporting

All my concerns have been amended by the authors.

Experimental design

All my concerns have been amended by the authors.

Validity of the findings

All my concerns have been amended by the authors.

Additional comments

No comment